# Design of Novel Fiber Optical Flexible Routing System

**Po-Tsung Wu and Tsair-Chun Liang \***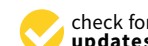

Institute of Photonics Engineering, National Kaohsiung University of Science and Technology, Kaohsiung City 824, Taiwan; e770103@gmail.com
* Correspondence: tcliang@nkust.edu.tw

**Abstract:** In this paper, we propose a new versatile routing device that utilizes arrayed waveguide gratings (AWGs), optical switches, and optical circulators to implement reconfigurable optical add/drop multiplexers (ROADMs), optical interleavers, and optical cross-connect (OXC). With the development of dense wavelength division multiplexing (DWDM) technology, ROADM and OXC technologies have also been put into practical use. Thus, the optical signal can be routed directly in the optical network according to its wavelength without the need for optical-electrical-optical (OEO) conversion. Although different optical network units (ONUs) have different bandwidth requirements, the use of optical interleavers has successfully solved the connection problem between old and new systems. According to the numerical experiments of static characteristics, the proposed routing device can effectively implement three different functionalities, thereby providing greater flexibility for fiber optic network applications.

**Keywords:** optical router; arrayed waveguide grating (AWG); optical cross-connect (OXC); reconfigurable optical add/drop multiplexer (ROADM); optical interleaver

## 1. Introduction

In today's era, there is an explosion of information and the amount of transmitted network data is growing at a rate of 7% to 20% per month. Because fiber-optic networks meet this high-capacity communication requirement, they are the backbone of present, and next-generation, high-speed networks. As the demand for wavelengths of optical communication signals continues to increase, the number of fibers laid in adjacent network nodes has also increased. Systems containing reconfigurable optical add/drop multiplexers (ROADMs) and optical cross-connects (OXCs) have generated a large number of development requirements [1–3]. Dense wavelength division multiplexing (DWDM) technology has also been developed. Each optical fiber can transmit hundreds of wavelength channels simultaneously, and each wavelength channel can carry several gigabits per second.

Due to the rapid development of the Internet of Things (IoT), network traffic will continue to increase, coupled with the popularity of mobile internet, and the continuous introduction of new 4 k/8 k high-definition audio and video services, these applications will increase the number of fibers required in each adjacent network node. Especially in densely populated metropolitan areas, network traffic is growing much larger than the core networks [4]. Therefore, the mutual application of ROADM and OXC has been widely discussed [5,6].

In addition, an optical component for expanding or contracting the number of each fiber channels is called an optical interleaver. Optical interleavers that can be used to separate or combine odd and even channels are also widely used in DWDM applications [7], which is one of the key components of the DWDM system [8,9]. The interleaver can be used to merge two sets of DWDM channels. In other words, reducing the channel spacing can double the number of optical signal channels per fiber. By using optical interleavers, existing DWDM systems can increase the amount of channels that can be transmitted, which can increase network utilization [10,11]. On the other hand, it can double the channel spacing of the original DWDM and allow the existing network system to connect smoothly to the new network system [12,13].

The optical network of metropolitan area networks (MANs) are mainly concentrated in metropolitan areas, and are used to connect the local central office (CO) and various network nodes. That is, to establish a high-speed broadband optical network in the most densely populated metropolitan areas. In addition, for multimedia applications, online shopping, video services, video calls, and video transmissions, it is more advantageous to use DWDM features to increase bandwidth for metropolitan areas that require relatively large bandwidth services. However, the main difference between the metropolitan area networks and backbone networks is that the DWDM systems of the long-distance backbone networks are all peer-to-peer network architectures, and the metropolitan DWDM network systems are a large number of data transmissions between the network nodes. Therefore, OXC and ROADM are the key technologies for signal wavelength switching between these optical network nodes.

The all optical fiber network has gradually matured in recent years, especially in backbone networks. DWDM technology has the function of effectively expanding the transmission capacity and distance of networks [14]. Although DWDM technology can effectively expand the transmission capacity of a single optical fiber, but it can only be used on long distance point-to-point architectures. When the network architecture is more complex, it requires the help of optical add/drop multiplexer (OADM) and OXC to complete the network construction [15]. However, to add or drop optical signal wavelengths in a traditional OADM, the system power budget must be calculated, resulting in an increase in cost, and an increased likelihood of occurrence of errors.

With the evolution of optical communication technology, the ROADM has matured over the last few years [16]. DWDM, ROADM, OXC, and optical interleavers can be used to automatically configure all signal wavelengths on ROADM nodes to increase backbone bandwidth or provide flexible bandwidth allocation techniques. DWDM technology can effectively expand the transmission capacity of a single optical fiber. ROADM and OXC can dynamically adjust or allocate network resources to meet the needs of dynamic networks. Because of their ability to dynamically adjust, we can avoid changing the network architecture in the event of future bandwidth shortages.

In DWDM optical networks, in order to upload/download the required signal wavelengths or to exchange the required signal wavelengths, this must be achieved by some common passive optical components, such as fiber Bragg grating (FBG), arrayed waveguide grating (AWG) [17–19], optical switches, optical circulators, and so on [20,21]. AWG is an important element commonly used to realize ROADM. In 2018, C. M. Tsai proposed a bi-directional ROADM based on a pair of N × N cyclic AWGs and tunable FBGs over N wavelength channel configurations [22]. Q. Huang et al., used several optical switches and a Mach–Zehnder interferometer to achieve the aim of adding or dropping multiple optical signal wavelengths at the same time [23].

Common OXC systems are mostly implemented using passive optical fiber components [24]. As far as we know, some published literature uses optical circulators and tunable FBGs to build OXC systems [25,26]. This architecture successfully completed the purpose of multi-signal wavelength switching. However, if the OXC function is performed using FBG, when the signal wavelength of the network increases, the number of FBGs must be increased, so the flexibility is insufficient. In recent years, the collocation of OXC and ROADM systems has been widely discussed [27–29]. It can be seen that these two systems are very important under the all-fiber network architecture.

In DWDM network, the combination of multi-functional systems are very important. Especially in next generation high-speed optical networks, the distribution or integration of optical signal wavelengths is an important task. In this study, we implemented a routing device with three different functionalities in the same router configuration, namely ROADM, OXC and a bidirectional interleaver. The proposed router can be used as a DWDM metropolitan area network, inter-domain switching medium (OXC function), and can help connect new and old systems (interleaver function) to increase network flexibility. Furthermore, we used optical switches to select the desired channel in this study, which is different from previous studies using the channels required for FBG modulation [3]. In addition to greatly improving the response speed of optical signal exchange, the system's homodyne crosstalk will also increase by about 30 dB when the ROADM is executed, which improves the performance of the system.

## 2. Configuration and Operating Principle of the Proposed Optical Routing System

The diagram of the proposed optical routing system is shown in Figure 1. In this configuration, two pairs of $8 \times 8$ AWGs with 100 GHz channel spacing, 11 $2 \times 2$ cross/bar optical switches (SWs), two optical couplers, and eight optical circulators (OCs) were used to construct the all optical routing system. Table 1 lists the spectral routing rule of $8 \times 8$ AWG. In the proposed router, we can adjust the status of optical switches SW1, SW2 and SW3 to implement the interleaver, OXC, or ROADM functions. The optical switch statuses of the various functions of the proposed optical routing system are shown in Table 2. Then we placed eight optical switches and corresponding signal wavelength channels on the right side of the AWG3 and AWG4 to determine the optical wavelength routing function that is allowed to pass through or be dropped. For optical switches numbered SW4 to SW11, when the optical switch is in the cross status, the signal will be dropped to drop port. If the optical switch is in the bar status, the signal will be passed to the output port.

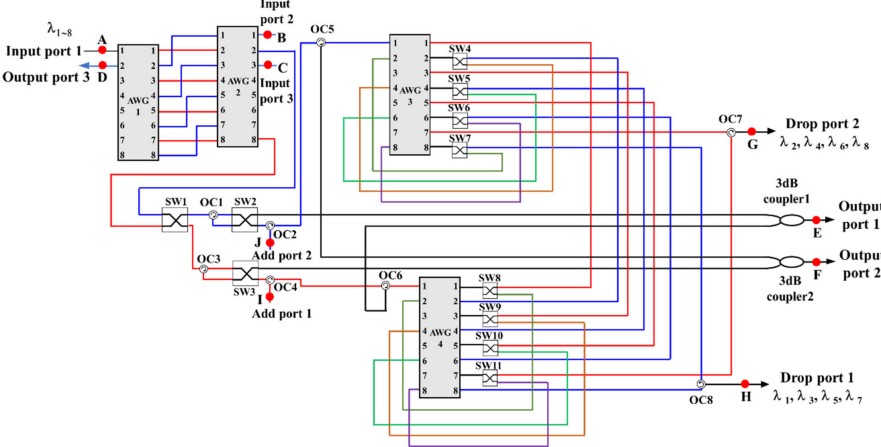

**Figure 1.** The configuration of proposed optical routing system. OC = optical circulators, AWG = arrayed waveguide gratings, SW = optical switches.

**Table 1.** $8 \times 8$ cyclic AWG routing rule.

| | | \multicolumn{8}{c}{Input Port} | | | | | | | |
| | | 1 | 2 | 3 | 4 | 5 | 6 | 7 | 8 |
|---|---|---|---|---|---|---|---|---|---|
| | 1 | $\lambda_1$ | $\lambda_2$ | $\lambda_3$ | $\lambda_4$ | $\lambda_5$ | $\lambda_6$ | $\lambda_7$ | $\lambda_8$ |
| | 2 | $\lambda_2$ | $\lambda_3$ | $\lambda_4$ | $\lambda_5$ | $\lambda_6$ | $\lambda_7$ | $\lambda_8$ | $\lambda_1$ |
| | 3 | $\lambda_3$ | $\lambda_4$ | $\lambda_5$ | $\lambda_6$ | $\lambda_7$ | $\lambda_8$ | $\lambda_1$ | $\lambda_2$ |
| Output Port | 4 | $\lambda_4$ | $\lambda_5$ | $\lambda_6$ | $\lambda_7$ | $\lambda_8$ | $\lambda_1$ | $\lambda_2$ | $\lambda_3$ |
| | 5 | $\lambda_5$ | $\lambda_6$ | $\lambda_7$ | $\lambda_8$ | $\lambda_1$ | $\lambda_2$ | $\lambda_3$ | $\lambda_4$ |
| | 6 | $\lambda_6$ | $\lambda_7$ | $\lambda_8$ | $\lambda_1$ | $\lambda_2$ | $\lambda_3$ | $\lambda_4$ | $\lambda_5$ |
| | 7 | $\lambda_7$ | $\lambda_8$ | $\lambda_1$ | $\lambda_2$ | $\lambda_3$ | $\lambda_4$ | $\lambda_5$ | $\lambda_6$ |
| | 8 | $\lambda_8$ | $\lambda_1$ | $\lambda_2$ | $\lambda_3$ | $\lambda_4$ | $\lambda_5$ | $\lambda_6$ | $\lambda_7$ |

**Table 2.** Optical switch statuses of the proposed optical routing system.

| Routing Function | \multicolumn{3}{c}{Switch Status} | | |
| | SW1 | SW2 | SW3 |
|---|---|---|---|
| Interleaver | Bar | Bar | Bar |
| OXC* | Cross | Bar | Bar |
| ROADM** | Bar | Cross | Cross |

*Optical cross-connect, **Reconfigurable optical add/drop multiplexers.

From Table 2, if we want the router to perform the functions of interleaver and de-interleaver, we must set SW1, SW2, and SW3 to the bar statuses. In Figure 1, when channel 1 ($\lambda_1$) to channel 8 ($\lambda_8$) are input from input port 1 of AWG1, the odd and even channels will come out from port 8 and port 2 on the right side of AWG2, respectively. Then, the odd channels will pass through SW1, OC3,

and SW3 to reach the output port 2, and the even channels will reach the output port 1 via SW1, OC1, and SW2 to implement the de-interleaver function. When we want to combine two sets of even and odd channels, we can input even and odd channels from input port 2 and input port 3 of the AWG2, respectively. These channels will be combined at output port 3 of AWG1 for interleaver functionality.

As can be seen from Table 2, when SW1 is in the cross status and SW2 and SW3 are in the bar statuses, we can perform the OXC function. Because the even channels from port 2 on the right side of AWG2 will reach the OC3 by the cross status of SW1, and then go to output port 2 from the bar status of SW3. Similarly, the signal input by input port 1 of AWG1, the odd channels will come out from port 8 on the right side of AWG2, then reach the OC1 by the cross status of SW1, and then reach the output port 1 through the bar status of SW2. That is, when SW1 is in the cross status and SW2 and SW3 are in the bar statuses, the even channels will be switched to output port 2 and the odd channels will be switched to output port 1.

In addition, when the SW1 was set to the bar status and SW2 and SW3 were set to the cross statuses, the proposed router will realize the ROADM functionality to drop or add certain signal wavelengths. For example, in this status setting, the odd channel $\lambda_1$ will appear from the port 1 on the right side of the AWG4 through SW1, OC3, SW3, OC4 and OC6 after input from the input port 1. Because we are dropping the wavelength $\lambda_1$, SW8 is set to the cross status. Therefore, the wavelength $\lambda_1$ comes out of port 1 on the right side of AWG4, then returns to port 2 on the left side, and then comes out from port 8 on the right side. Finally, wavelength $\lambda_1$ reaches drop port 1 via OC8. When $\lambda_1$ is dropped, we can reuse the $\lambda_1$ carrier to add another signal from add port 1. Since the optical switch SW3 is in the cross status, when $\lambda_1$ is added from OC4, it passes through SW3 and OC3 and then returns to SW3, and then to the output port 2. Similarly, we can drop the even-channel from drop port 2 and add it from add port 2.

As mentioned above, the proposed all-optical router can perform the functions of an interleaver through the routing rules of the AWGs. The functions of OXC and ROADM can also be achieved by AWGs and optical switches. Therefore, this designed router can be directly applied to the DWDM network system.

## 3. Numerical Experiments and Results

In this study, we used OptiSystem software [30] to verify and evaluate the performance of the proposed optical routing system. OptiSystem is an optical communication system simulation software package for designing, testing and optimizing virtually any type of optical link in a wide range of optical network physical layers, from analog video broadcast systems to intercontinental backbone networks. The experimental setup is shown in Figure 2, in which the block diagram of the proposed routing system is shown in Figure 1. The red connection points of A, B, C, D, and so on, in Figure 2, correspond to the connection points in Figure 1. During the simulation, we used optical spectrum analyzers (OSAs) to measure static performance to verify the feasibility of its function and evaluate the performance of the optical interleaver, OXC, and ROADM functionalities.

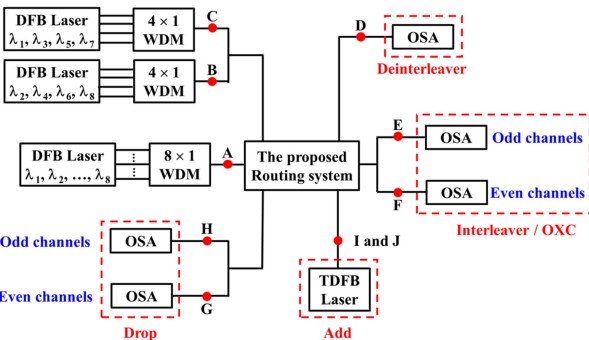

**Figure 2.** Experimental setup for the demonstration of the proposed optical routing system.

In this study, the insertion loss of the C-band of the various components of the proposed router, AWG, WDM, SW and OC were set to 2.2 dB, 1.5 dB, 1 dB and 0.7 dB, respectively. We used eight

distributed feedback (DFB) lasers as eight independent channels with an input power of −10 dBm per channel. The channel spacing in the experiment was 0.8 nm (100 GHz), depending on the channel spacing of the AWG used. The wavelength of the signal belongs to the C-band and ranges from 1547.6 nm to 1553.2 nm. At present, all optical switches sold on the market have switching speeds of less than 10 milliseconds. Therefore, we used it as a research benchmark for future experiments to test dynamic performance. Since the OptiSystem software does not have the function of setting the switching speed, the switching time of this static performance study was set to the default value of zero by the software. Figure 3 shows the spectrum of the eight DFB laser sources.

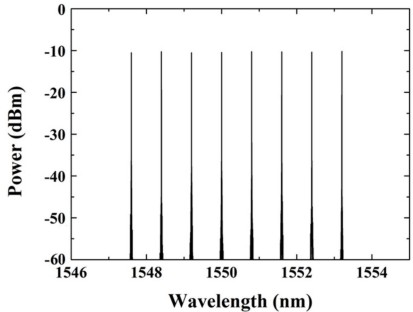

**Figure 3.** The spectrum of eight distributed feedback (DFB) laser sources.

### 3.1. Static Spectral Performance of Interleaver

In the proposed router architecture, when the statuses of SW1, SW2 and SW3 are set to bar, the functions of the interleaver and de-interleaver can be implemented by AWG routing rules. Figures 4 and 5 show the spectrum of the even and odd channels of output port 1 and output port 2 after de-interleaver processing. The even channels are $\lambda_2$ = 1548.4 nm, $\lambda_4$ = 1550.0 nm, $\lambda_6$ = 1551.6 nm, and $\lambda_8$ = 1553.2 nm. The odd channel wavelengths are $\lambda_1$ = 1547.6 nm, $\lambda_3$ = 1549.2 nm, $\lambda_5$ = 1550.8 nm, and $\lambda_7$ = 1552.4 nm. From Figure 1, when we input even and odd channels from the ports of input two and input three, respectively, the even and odd channels were combined and output to output port 3. The result is shown in Figure 6.

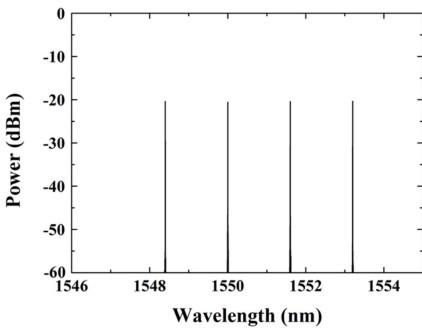

**Figure 4.** The spectrum of even signal wavelengths at output port 1 using the de-interleaver function.

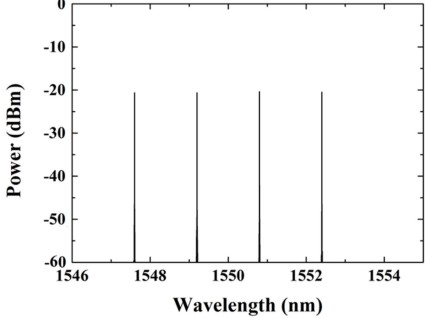

**Figure 5.** The spectrum of odd signal wavelengths at the output port 2 using the de-interleaver function.

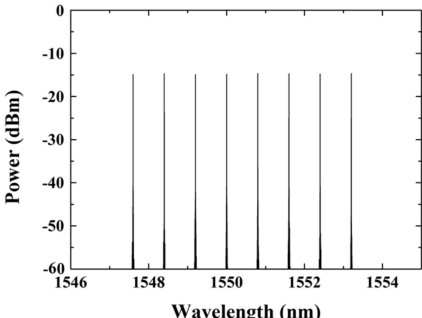

**Figure 6.** The spectrum of the combination of odd and even channels at output port 3 using the interleaver function.

## 3.2. Static Spectral Performance of OXC

Optical cross-connect (OXC) is a device used to switch high-speed optical signals in a fiber optic network. In Figure 1, when SW1 is in the cross status and SW2 and SW3 are in the bar statuses, we can use this router to exchange the odd and even channels on the output port. That is, the odd channel wavelengths output at output port 2 will be switched to output port 1, and the even signal wavelengths output at output port 1 will be switched to output port 2. Figures 7 and 8 show the spectrum of the output signals at output port 1 and output port 2 when the function of OXC was performed.

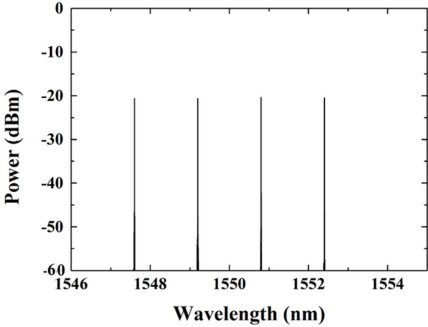

**Figure 7.** The spectrum of odd signals at the output port 1 realizing OXC functionality.

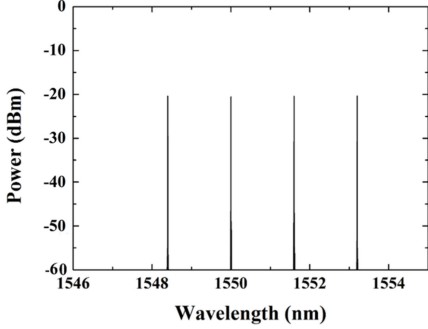

**Figure 8.** The spectrum of even signals at the output port 2 realizing OXC functionality.

## 3.3. Static Spectral Performance of ROADM

In the ROADM system, we selected the signal wavelengths $\lambda_2$ and $\lambda_5$ to verify the add/drop function in the proposed routing system. When the state of SW1 is set to the bar status, and the states of SW2, SW3, and the optical switches SW4 and SW10 corresponding to the two selected channels are set to the cross statuses, the ROADM function will be performed. The two selected channels will be routed to the drop port during the drop process. Figures 9–12 show the simulated spectrum at the output port and drop port with the drop process.

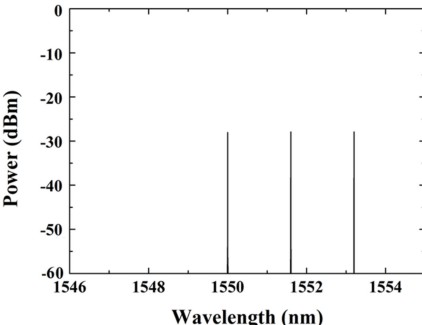

**Figure 9.** The spectrum of the even channels at the output port 1 with the drop process.

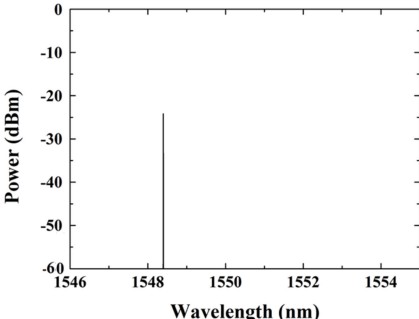

**Figure 10.** The spectrum of $\lambda_2$ at the drop port 2 with the drop process.

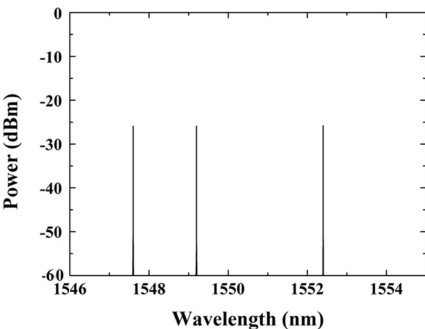

**Figure 11.** The spectrum of the odd channels at the output port 2 with the drop process.

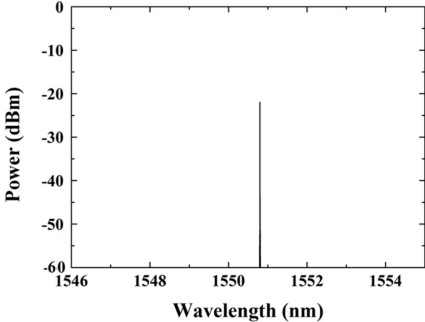

**Figure 12.** The spectrum of $\lambda_5$ at the drop port 1 with the drop process.

While dropping the signal, we can also add the optical signals of the same carrier as the drop signals (that is, the signal $\lambda_2$ and $\lambda_5$) to add port 1 and add port 2. All added channel input power is also −10 dBm. Since SW2 and SW3 are set to the cross statuses, these added signal wavelengths will be routed to output port 1 and output port 2. Figures 13 and 14 show the spectrum at the output port 1 and port 2 with the add process.

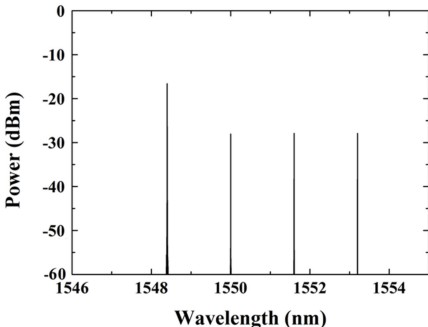

**Figure 13.** The spectrum at the output port 1 with the add $\lambda_2$ process (even channel).

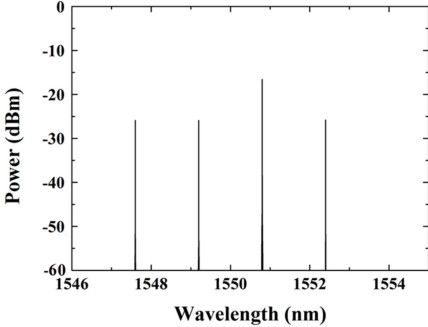

**Figure 14.** The spectrum at the output port 2 with the add $\lambda_5$ process (odd channel).

## 4. Conclusions

In this research, an all-optical routing system based on AWGs, optical circulators, couplers and optical switches was used to implement a bidirectional interleaver, OXC, and ROADM with three different functionalities. In the case of the bidirectional interleaver, the even and odd channels implement multiplexer and demultiplexer functions through interleaver processing based on AWG routing rules and the status of the corresponding SWs. The proposed routing system was shown to effectively implement the bidirectional interleaver function. For the proposed routing system, the OXC function can also be tested by using the input odd and even channels. That is, the odd and even channels are exchanged at the output port. In the ROADM verification process, we realized the function of adding and dropping through the bar or cross status of optical switches. We used two optical signal wavelengths (i.e., $\lambda_2$ and $\lambda_5$) to verify the feasibility of the proposed routing device with ROADM functionality. Based on the numerical experiments of static characteristics, the proposed routing device can effectively implement the routing functions of an interleaver, OXC, and ROADM.

After simulation analysis, we have determined the feasibility of the proposed router architecture, and then we will conduct actual optical path experiments, including static and dynamic characteristics, to test the performance of the proposed router. In addition, the AWG has a wavelength cycling characteristic of light, so more than eight signal wavelengths can be configured in the proposed routing system. Furthermore, the choice of input signal channel spacing depends entirely on the characteristics of the AWG used. The AWG channel spacing used in this research system is 100 GHz, so the input signal channel uses a 100 GHz spacing. We can change the channel spacing of the input wavelength by using different channel spacing AWGs.

The biggest difference between our proposed router system and available systems is that we integrate the three functions of ROADMs, an optical interleaver, and OXC, and there is no need to add additional optical components when switching between functions. To the best of our knowledge, currently available products do not have an optical router with these three functions. This investigation provides a more flexible router system for DWDM networks.

**Author Contributions:** Conceptualization, P.-T.W. and T.-C.L.; investigation, P.-T.W. and T.-C.L.; Experiments, P.-T.W.; validation, T.-C.L.; funding acquisition, P.-T.W. and T.-C.L.; project administration, T.-C.L.; writing—original draft preparation, P.-T.W.; writing—review and editing, T.-C.L.

**Funding:** This research was funded by the Ministry of Science and Technology of Taiwan, grant number MOST 105-2221-E-327-028.

**Acknowledgments:** The authors would like to thank Wei-Ning Shih for drawing the optical routing system configuration diagram and experimental results presented in this paper.

**Conflicts of Interest:** The authors declare no conflict of interest.

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
