# Peer review of "Design of Novel Fiber Optical Flexible Routing System"

_applsci, doi:10.3390/app9224763_

Round 1

Reviewer 1 Report

The paper has the merit that it fully describes the structure and operation of a novel optical router (incorporating three routing functions) and provides a good simulation of its performance.

Some remarks/suggestions:

The authors should give more emphasis in how the proposed system differentiates itself from the available systems as well as its advantages.

The "Conclusions" section should be a bit more detailed. For example, it could include a discussion on possible future research and/or implementation issues.

The "OptiSystem" software should be mentioned in the reference list.

Moderate changes in the use of English (and/or rephrasing, in some cases) are necessary (see, for example, lines 41-42, 50-51, 66-68, 84-85 to indicate a few).

An optional comment: Though the simulation does not provide information on the switching speed, is there any clue on the possible speed of the proposed configuration? 

Author Response

Response to Reviewer 1

[Reviewer comments]:

Comments and Suggestions for Authors

The paper has the merit that it fully describes the structure and operation of a novel optical router (incorporating three routing functions) and provides a good simulation of its performance.

Some remarks/suggestions:

(1) The authors should give more emphasis in how the proposed system differentiates itself from the available systems as well as its advantages.

(2) The "Conclusions" section should be a bit more detailed. For example, it could include a discussion on possible future research and/or implementation issues.

(3) The "OptiSystem" software should be mentioned in the reference list.

(4) Moderate changes in the use of English (and/or rephrasing, in some cases) are necessary (see, for example, lines 41-42, 50-51, 66-68, 84-85 to indicate a few).

(5) An optional comment: Though the simulation does not provide information on the switching speed, is there any clue on the possible speed of the proposed configuration?

[Author responses]:

(1) The biggest difference between our proposed router system and the available system is that we integrate the three functions of ROADMs, optical interleaver, and OXC, and there is no need to add additional optical components when switching between functions. As far as we know, currently available products do not have optical routers with these three functions at the same time. We have emphasized these differences in the Conclusions. In addition, we used optical switches to select the desired channel in this study, which can greatly improve the response speed of optical signal exchange, which is different from previous studies using FBG modulation required channels. We have already described it in the Introduction.

(2) We have improved the Conclusions seriously according to the reviewer’s comments. Thanks for the reviewer’s kindly comments. Another, we have now completed the proposed router feasibility assessment analysis. The next step is to measure the actual optical path, including static and dynamic characteristics, to actually test the proposed router performance. We have added this to the Conclusions.

(3) OptiSystem is an optical communication system design software designed by Optiwave Systems, a software company based in Ottawa. Based on the reviewer's comments, we have added references [30] to the revised manuscript, which is available on the Optiwave Software website.

(4) We have carried out thorough check for our paper and make necessary modifications. Thank you for your kindly and helpful suggestion.

(5) Thanks for the reviewer’s thoughtful comments. After simulation analysis, we have determined the feasibility of the proposed router architecture, and then we will conduct actual optical path experiments. At present, the switching time of commercially available optical switches is mostly less than 10 ms, and the switching time of the optical switches we have purchased are less than or equal to 8 ms.

Reviewer 2 Report

General, the paper is well written, and the covered topic in quite interesting. The way the authors present their work is in the right order and allows the reader to follow everything step by step. Achieved results are also interesting and enable the implementation of one device with three different functionalities.

Some suggestions are as follows:
1) In the caption of Figures 7 and 8, I recommend changing "... port 1 (2) with doing OXC function" into "...port 1 (2) realizing OXC functionality".

2) It is better to use the "spectrum" instead of the "spectra."

Author Response

Response to Reviewer 2

[Reviewer comments]:

Comments and Suggestions for Authors

General, the paper is well written, and the covered topic in quite interesting. The way the authors present their work is in the right order and allows the reader to follow everything step by step. Achieved results are also interesting and enable the implementation of one device with three different functionalities.

Some suggestions are as follows:

(1) In the caption of Figures 7 and 8, I recommend changing "... port 1 (2) with doing OXC function" into "...port 1 (2) realizing OXC functionality".

(2) It is better to use the "spectrum" instead of the "spectra."

[Author responses]:

(1) We have made changes based on the reviewer's recommendations. Thank you for your kindly suggestion.

(2) We have changed "spectra" to "spectrum". Thanks for the reviewer’s comments.
